# Lipid Metabolism and Epigenetics Crosstalk in Prostate Cancer

**DOI:** 10.3390/nu14040851

**Published:** 2022-02-18

**Authors:** Juan C. Pardo, Vicenç Ruiz de Porras, Joan Gil, Albert Font, Manel Puig-Domingo, Mireia Jordà

**Affiliations:** 1Department of Medical Oncology, Catalan Institute of Oncology, University Hospital Germans Trias i Pujol, Ctra. Can Ruti-Camí de les Escoles s/n, 08916 Badalona, Spain; jcpardor@iconcologia.net (J.C.P.); afont@iconcologia.net (A.F.); 2Catalan Institute of Oncology, Badalona Applied Research Group in Oncology (B·ARGO), Ctra. Can Ruti-Camí de les Escoles s/n, 08916 Badalona, Spain; vruiz@igtp.cat; 3Germans Trias i Pujol Research Institute (IGTP), Ctra. Can Ruti-Camí de les Escoles s/n, 08916 Badalona, Spain; jgil@igtp.cat (J.G.); mpuigd@igtp.cat (M.P.-D.); 4Department of Endocrinology and Medicine, CIBERER U747, ISCIII, Research Center for Pituitary Diseases, Hospital Sant Pau, IIB-SPau, Universitat Autònoma de Barcelona, 08041 Barcelona, Spain; 5Department of Endocrinology and Nutrition, University Germans Trias i Pujol Hospital, Ctra. Can Ruti-Camí de les Escoles s/n, 08916 Badalona, Spain; 6Department of Medicine, Autonomous University of Barcelona (UAB), Ctra. Can Ruti-Camí de les Escoles s/n, 08916 Badalona, Spain

**Keywords:** prostate cancer, lipid metabolism, epigenetics, diet, fatty acid, cholesterol, DNA methylation, histone modifications, predictive biomarkers, therapeutic vulnerabilities

## Abstract

Prostate cancer (PCa) is the most commonly diagnosed malignant neoplasm in men in the Western world. Localized low-risk PCa has an excellent prognosis thanks to effective local treatments; however, despite the incorporation of new therapeutic strategies, metastatic PCa remains incurable mainly due to disease heterogeneity and the development of resistance to therapy. The mechanisms underlying PCa progression and therapy resistance are multiple and include metabolic reprogramming, especially in relation to lipid metabolism, as well as epigenetic remodelling, both of which enable cancer cells to adapt to dynamic changes in the tumour. Interestingly, metabolism and epigenetics are interconnected. Metabolism can regulate epigenetics through the direct influence of metabolites on epigenetic processes, while epigenetics can control metabolism by directly or indirectly regulating the expression of metabolic genes. Moreover, epidemiological studies suggest an association between a high-fat diet, which can alter the availability of metabolites, and PCa progression. Here, we review the alterations of lipid metabolism and epigenetics in PCa, before focusing on the mechanisms that connect them. We also discuss the influence of diet in this scenario. This information may help to identify prognostic and predictive biomarkers as well as targetable vulnerabilities.

## 1. Introduction

Prostate cancer (PCa) is the most commonly diagnosed malignant neoplasm and the second leading cause of cancer-related death in men in the Western world [1]. There is an increasingly evident relationship between PCa and the state of chronic inflammation associated with obesity [2]. Importantly, in the coming decades, due to the progressive aging of the population, the high incidence of obesity, and the Western dietary habits common in developed countries, the incidence of PCa will increase significantly, which will represent an important public health problem.

About 80–90% of PCa patients are diagnosed at localised or locally advanced stages, and most of them can be cured with local treatments, such as surgery or radiotherapy, with or without androgen deprivation therapy (ADT) [3]. However, about 10% of patients present metastases at initial diagnosis. These patients, termed de novo metastatic castration-sensitive PCa (mCSPC) patients, have a shorter overall survival (OS) compared with patients who develop metastases years after the initial diagnosis [4,5]. Importantly, despite the current use of the prostate-specific antigen (PSA) test as a screening method, the number of patients who will present with mCSPC is expected to increase in the coming years.

It is well known that PCa is an androgen-dependent disease and activation of the androgen receptor (AR) is essential for tumour progression [6]. Hence, for patients with PCa who experience disease relapse after local therapy or for those with de novo mCSPC, ADT is the backbone of systemic therapy. However, despite significant initial responses to ADT, almost all metastatic patients progress to an incurable metastatic castration-resistant PCa (mCRPC), defined as radiographic progression and/or a rise in PSA levels despite having a castrate level of testosterone [7,8]. mCRPC has a poor prognosis, with a median OS of approximately three years, and poses a major therapeutic challenge. Since 2004, docetaxel-based chemotherapy has been the standard first-line treatment for mCRPC [4,9]. Moreover, the armamentarium of treatments for PCa has increased in recent years thanks to the FDA/EMA approval of new AR pathway inhibitors (ARPIs) such as abiraterone acetate [10,11,12], an irreversible inhibitor of the cytochrome P450 17α−hydroxy/17,20-lyase (CYP17) enzyme responsible for androgen synthesis, and next-generation anti-androgens such as enzalutamide [13,14,15,16], apalutamide [17,18], and darolutamide [19], which have changed the therapeutic landscape of PCa at different stages of the disease. However, 10–17% of mCRPC patients may develop an aggressive variant of PCa (AVPCa) such as neuroendocrine PCa (NEPC), which is associated with aggressive clinical features and poor prognosis [20]. The risk of developing NEPC is increased in patients with prolonged exposure to ARPIs.

Although the therapeutic landscape of mCRPC has changed drastically in the last decade with the incorporation of new therapeutic strategies, mCRPC still has a poor prognosis, mainly due to disease heterogeneity and the development of resistance to therapy. Therefore, there is an urgent yet unmet clinical need to find new predictive biomarkers to guide the best sequence of systemic therapy and to identify new therapeutic targets. It is now more necessary than ever to understand the molecular landscape of PCa.

Several studies have shown that epigenetic changes occur during the earliest stages of PCa, suggesting they are an essential step in PCa initiation, and are maintained throughout disease progression and metastasis [21,22]. Strong evidence also indicates that the evolution to AVPCa, like NEPC, is driven by coordinated epigenetic and transcriptional reprogramming [23]. In fact, epigenetic mechanisms, particularly DNA methylation and post-translational histone modifications, have a well-established role in regulating cell plasticity through the alteration of chromatin structure and DNA accessibility, thereby influencing gene expression programs and ultimately the phenotype [24,25].

Epigenetic alterations are a hallmark of human cancers [26,27]. In contrast to the genome, the epigenome is less stable and can change during cell cycle or in response to different stimuli, including external factors, such as physical activity, pollution, or diet [28]. Epigenetics can thus act as a link between the environment and the tumour phenotype. In several cancers, including PCa, diet can induce epigenetic modifications that result in global alterations of chromatin packaging, thereby modulating gene expression profiles and altering the course of the disease [29]. It was reported that a high-fat diet (HFD) enhanced tumour progression of human PCa in mouse models [30,31] and was also linked to increased neuroendocrine differentiation [32]. Obesity, which is related to a HFD, has also been linked to an elevated risk of developing AVPCa. However, the relationship between obesity and PCa is a complex phenomenon involving multiple biological factors, including variation in sex hormone levels and adipokines, systemic inflammation, alterations in the insulin and IGF-1 axis, and metabolic alterations [33,34].

Not only uptake of lipids from the circulation, but also de novo lipid synthesis, partly stimulated by androgens, play important roles in PCa and are correlated with tumour progression and poor prognosis [35,36]. In fact, aberrant metabolism is a hallmark of cancer, as tumour cells need to rewire their metabolism to satisfy the energy requirements and biochemical needs stemming from their uncontrolled growth and other oncogenic capacities [37]. This can lead the way towards new opportunities for PCa treatment by targeting enzymes associated with de novo lipogenesis pathways.

In the present review, we focus on the current knowledge of the link between lipid metabolism and epigenetics, specifically DNA methylation and histone modifications, in PCa. We first present an overview of the alterations of lipid metabolism and epigenetics in PCa and highlight those with potential as prognostic and predictive biomarkers and/or therapeutic targets. Next, we discuss the potential effect of diet, in particular fats, on PCa, lipid metabolism, and epigenetics. Finally, we discuss how metabolism influences epigenetics by altering the availability of metabolites, specifically acetyl-CoA, non-acetyl acyl-CoAs, NAD+, and SAM, that are used by epigenetic enzymes, and how epigenetics regulates the expression of metabolic enzymes. Finally, we explore future perspectives in PCa.

## 2. Lipid Metabolism Alterations: A Hallmark of PCa

Cellular metabolism consists of a complex network of enzymes, substrates, and signalling pathways involved in multiple, tightly controlled, cellular processes that maintain homeostasis and adapt to normal physiological changes, such as those produced by circadian rhythm, nutrient intake, or physical activity [38]. During tumour progression, infiltration, metastasis, adaptation and clonal selection, tumour cells rewire their metabolism, which allows them to cope with the stress produced by hypoxia and lack of nutrients in the microenvironment [39]. However, although the dysregulation of metabolism was included in the updated hallmarks of cancer by Hannahan and Weinberg in 2011 [40], few advances have made an impact on clinical practice.

For years, glucose metabolism has been highly explored. Glucose metabolism is commonly altered in many types of tumours in which the rate of glucose uptake dramatically increases and lactate is produced, even in the presence of oxygen. This process, called the Warburg effect or aerobic glycolysis, is an inefficient catabolic process compared to oxidative phosphorylation (OXPHOS) [41,42]. New insights have concluded that aerobic glycolysis is not necessarily due to a common aberrant mitochondrial function but is related to the cell type of origin and alterations in the microenvironment due to the limited availability of substrates and oxygen [43].

Normal prostate metabolism is unique and characterized by an altered tricarboxylic acid (TCA) cycle due to the high accumulation of zinc in the prostate gland (Figure 1). This inhibits the enzyme aconitase 2 (ACO2), which is responsible for catalysing the isomerization of citrate in the TCA cycle. This, in turn, promotes the accumulation of citrate to be secreted into the prostatic fluid and the demand to generate citrate is compensated by aspartate [44]. As a result, in normal prostate cells, the rate of OXPHOS is low, while aerobic glycolysis is high. Many of the components involved in the metabolic program of the prostate are regulated and coordinated by AR [45].

PCa, like most tumours, undergoes a metabolic shift (Figure 1), but in contrast to other tumours, PCa favours enhanced OXPHOS but limited glycolysis, ruling out the use of advanced diagnostic procedures such as standard fluorodeoxyglucose-positron emission tomography (FDG-PET) for the detection of tumour relapse in the early stages when PSA values are low [46,47]. This metabolic reprogramming is largely mediated by AR and the loss of zinc transporters during the oncogenic process [45,48]. Zinc depletion reactivates ACO2 and restores the TCA cycle, which is mainly fuelled by extracellular and intracellular pyruvate (Figure 2) [49]. Citrate is not accumulated but is oxidized in the TCA cycle and exported to the cytoplasm to generate acetyl-CoA as building blocks for fatty acids (FAs).

However, during progression to mCRPC there is a switch to the Warburg effect [50] (Figure 1). The Warburg effect seems to be promoted by bone marrow adipocytes, which activate the expression of glycolytic enzymes in mCRPC cells. In addition, increased lactate secretion into the tumour microenvironment is also associated with tumour aggressiveness.

Interestingly, specific genetic backgrounds under specific environmental conditions can induce a different metabolic landscape in PCa. This is the case for some germline NK3 Homeobox *1*
*(NKX3.1*) polymorphisms under oxidative stress [51]. The NKX3.1 cancer-associated variants R52C or T64A, present in 14% and 11% of the population, respectively, impair the mitochondrial function of NKX3.1 and lead to the dysfunction of mitochondrial OXPHOS and promotion of aerobic glycolysis which can result in a more aggressive early PCa phenotype. These findings open new avenues to target subsets of men at increased risk of developing PCa or subsets of PCa patients with specific molecular and metabolic phenotypes. In line with this, the use of prophylactic mitochondrial antioxidant therapy may benefit individuals harbouring these *NKX3.1* polymorphisms. Moreover, as NKX3.1 expression and localization are associated with clinical outcome, their analysis may improve risk assessment in PCa patients, especially for men under active surveillance.

PCa, both at early and late stages, is also characterized by an increased de novo lipogenesis and FA oxidation (FAO) (Figure 1) through the upregulation of AR-regulated lipogenic enzymes to fulfil the energetic and anabolic needs of cancer cells [50]. In this setting, the transcription factors sterol regulatory element-binding proteins (SREBPs), which comprise three isoforms, SREBP1a, SREBP1c, and SREBP2, are the main regulators of lipogenesis [52]. The interaction between androgens and SREBPs is complex and mediated by AR, which enhances both the expression and the activation of SREBPs. AR can also directly regulate the expression of lipogenic genes due to the proximity of AR binding sites (ARBS) to lipid metabolism gene promoters. It is interesting to note that there is a reciprocal relationship between these two pathways, as SREBP can activate the AR pathway by activating AR gene expression.

In addition to AR-dependent metabolic alterations, the PCa-associated genetic aberrations that are not related to AR, such as p53 loss, phosphatidylinositol-3-kinase (PI3K) mutations, and loss of Phosphatase and tensin homolog (PTEN), also enhance lipid metabolism, exemplifying the close relationship between genetic drivers and metabolic reprogramming. Mutants of the tumour suppressor p53 were reported to activate the SREBP-mediated metabolic pathways in metastatic PCa cells [53]. Several studies have shown that SREBP is also activated through the PI3K/Protein Kinase B/mammalian target of rapamycin (PI3K/AKT/mTORC1) oncogenic signalling pathway [54]. Finally, the co-deletion of PTEN and promyelocyticleukaemia protein (PML) also results in an SREBP-dependent activation of lipogenesis promoting metastatic PCa [55].

Overall, these data indicate that, unlike other tumours, prostate tumours are highly dependent on lipid metabolism (Figure 1). Consequently, lipids, including FAs, cholesterol, and phospholipids, play a key role in the development and progression of PCa, as summarized in the following sections.

### 2.1. Alterations of FA Metabolism in PCa

FAs are important energy sources, common components of complex lipids, and building blocks in anabolic processes [56]. Most normal cells can uptake dietary lipids from the circulation and can also synthesize de novo lipids using acetyl-CoA as building blocks. FA metabolism is therefore a complex network of interacting pathways ranging from uptake of external FAs, de novo synthesis, expression of regulatory genes, energy production through FAO, storage in lipid droplets (LD), and release of FAs from intracellular stores and surrounding tissues.

#### 2.1.1. De Novo FA Synthesis

PCa is characterized by an increased de novo FA synthesis that has been associated with energy storage, redox balance, increased requirement by cell membranes, and regulation of many key cellular processes that are involved in cell proliferation, apoptosis, differentiation, angiogenesis, inflammation, motility, epithelial–mesenchymal transition, and drug resistance [36,57]. FA synthesis occurs in the cytosol and the first reaction is the conversion of citrate, derived from the mitochondrial TCA cycle, to acetyl-CoA by ATP citrate lyase (ACLY) [57]. Acetyl-CoA is then converted to malonyl-CoA by the rate-limiting enzyme acetyl-CoA carboxylase (ACACA, commonly referred to as ACC). Subsequently, FASN catalyses the synthesis of palmitate, a saturated FA (SFA), which undergoes further modifications, specifically desaturation by fatty acyl-CoA desaturases (FADS) or stearoyl-CoA desaturases (SCDs), and elongation by elongases (ELOVLs), to generate monounsaturated FAs (MUFAs) and polyunsaturated FAs (PUFAs) (Figure 2).

Some of these enzymes are regulated by androgens and are overexpressed in PCa [35]. They play an important role in the development and progression of PCa and, importantly, have been proposed as biomarkers for prognosis. For example, FASN is upregulated in early PCa, increases during progression, and correlates with poor prognosis and survival [58]. Additionally, SCD1 overexpression correlates with a higher Gleason grade [59], and ELOVL7 is involved in PCa growth and survival [60]. Moreover, some of these enzymes can be targeted, making them a clinically exploitable vulnerability in the setting of mCRPC, and efforts are being made to develop specific inhibitors. In particular, FASN inhibition has been shown to antagonize the growth of mCRPC in in vitro models through metabolic reprogramming [61], and ACLY inhibition sensitizes mCRPC cells to enzalutamide, although no ACLY inhibitors have yet reached the clinical level [62]. There is also great interest in using dietary interventions to improve the efficacy of existing cancer treatments [63].

#### 2.1.2. FA Uptake

Although de novo FA synthesis is a key process of adaptation in PCa, exogenous FAs are also important and are regulated by AR signalling in different ways [64]. The major proteins involved in FA transport that are active in PCa include the FA translocase CD36, the plasma membrane FA binding protein (FABPpm), the cytosolic FABPs, and the FA transport protein (FATP) (Figure 2) [36]. CD36 is an FA receptor expressed in a broad variety of cell types and involved in many processes such as angiogenesis, atherosclerosis, inflammation, and lipid metabolism [65]. The uptake of FAs, such as palmitic acid, by CD36 has been associated with migration and metastasis in in vivo models of squamous cell carcinoma [66,67] and gastric cancer [68], yet it has not been investigated if it also occurs in PCa. Importantly, the inhibition of FA uptake by blocking CD36 has been associated with significant benefit in preclinical models of PCa, specifically a reduction in tumour growth and PCa severity. However, CD36 blockade can activate de novo lipogenesis, so combined blockade of both pathways through CD36 and FASN inhibition may provide greater antitumour activity than either of the single treatments alone [69].

FABPpm is located in the outer plasma membrane and promote cellular FA uptake [70], while FABPs are cytosolic and are involved in the intracellular trafficking of FAs by regulating its cellular accumulation, utilization, and fate [71]. PCa expresses different types of FABPs and their pattern of expression is altered when compared with normal prostate tissue [72]. Interestingly, androgens stimulate lipolysis of FAs from adipocytes [73] and can induce the expression of FABPs in PCa cells, which results in a paracrine cycle of FA release from the adipose-rich tumour environment and optimized FA uptake by tumour cells [74]. The volume and composition of the adipose tissue is related to diet, intake patterns, hormonal stimulation, genetics, and metabolic syndrome [75]. In this regard, ADT, the cornerstone of PCa treatment, promotes metabolic syndrome, which may influence lipid availability and consumption [76]. Moreover, FABPs have been related to prognosis in PCa [77,78]. Targeting this family of proteins in in vitro and in vivo models has shown a synergistic effect on PCa growth, specifically when combining FABP5 inhibitors with docetaxel/cabazitaxel [79].

Regarding FATPs, their expression is highly heterogeneous in PCa and their role and clinical implications are not fully clarified [36].

#### 2.1.3. FA Oxidation (FAO)

FAO is a multi-step process that occurs in the mitochondrion and in the perixosome. Mitochondrial FAO consists of the conversion of long chain FAs (LCFAs) into acetyl-CoA to generate energy while perixosomal FAO does not contribute to energy production directly but is required for the initial oxidation of very LCFAs (VLCFAs), branched chain FAs, and other FA derivatives that cannot be directly oxidized by the mitochondrion [80]. FAO is not only an important process to obtain energy but it has also been proposed as a gatekeeper that is dysregulated by oncogenic signals to drive tumorigenesis [81]. The overexpression of enzymes involved in FAO has been detected in various types of cancer, including PCa. LCFAs enter the cell, are converted into fatty acyl-CoA by the long chain acyl-CoA synthetase (ACSL), and then are shuttled into the mitochondria by the carnitine shuttle system, which consists of three enzymes (carnitine palmitoyltransferase (CPT) 1, carnitine/acylcarnitine translocase (CACT), and CPT2) (Figure 2). Once in the mitochondria, they enter the oxidation process. Metabolomic and transcriptomic data have revealed an elevated expression of CPT1 and CPT2 in PCa [82]. CPT1 is also amplified in 22% of NEPC [23]. Recent preclinical studies have shown that inhibition of lipid catabolism by blocking FAO results in increased activity of the AR pathway, leading to increased sensitivity to enzalutamide and reduced PCa growth [83]. 2,4-Dienoyl-CoA Reductase 1 (DECR1) is another enzyme involved in FAO, specifically in an auxiliary pathway for PUFA β-oxidation. Data from The Cancer Genome Atlas (TCGA) have revealed an increased expression of DECR1 in PCa that correlates with a higher Gleason score, more advanced disease, and shorter survival. In addition, in in vitro and in vivo models, there was a relationship between androgen signalling and inhibition of DECR1, and treatment with castration therapy and ARPIs produced an overexpression of DECR1 [84].

#### 2.1.4. FA Storage and Release

LDs are highly conserved cellular structures that store fat in the form of neutral lipids; they are composed of a core of triglycerides (TAGs) and hydrophobic sterol esters and surrounded by a hydrophobic phospholipid monolayer [85]. LDs are formed in response to various stimuli, including excess of lipids or a variety of stresses, and have diverse cellular functions, including mobilizing FAs to synthesize membranes or to obtain energy, lipid signalling pathways, and sequestering toxic lipids [86].

The accumulation of LDs has been observed in many cancers and is elevated in cancer cells exposed to hypoxia or nutrient starvation [87]. In PCa, androgens induce a marked stimulation of the synthesis and accumulation of neutral lipids in LDs [88]. Diacylglycerol acyltransferase 1 (DGAT1) is one of the major enzymes in TAG biosynthesis that specifically catalyses the final step of lipogenesis, converting diacylglycerol (DAG) and acyl-CoA to TAG (Figure 2). DGAT1 is overexpressed in PCa compared to normal prostate epithelium. The inhibition of this lipogenic enzyme reduced proliferation, migration and invasion of PCa cells in vitro and reduced PCa cell growth in vivo [89].

The best-characterized mechanism by which FAs are released from LDs involves cytoplasmic lipases. In response to various cellular signals, lipolysis is initiated by the enzyme adipose triglyceride lipase (ATGL) [90]. Unlike other cancer types, the role of ATGL in PCa is ambiguous based on the contradictory reported findings [91,92]. The action of ATGL is followed by hormonesensitive lipase (HSL), whose expression can be inhibited by androgens, as shown in human subcutaneous adipose tissue [93]. In PCa, HSL seems to be directly involved in the pathobiology of CRPC by triggering intratumoral de novo steroid synthesis which activates AR [94]. The final step in the lipolytic cascade is performed by monoacylglycerol lipase (MGL), which is highly expressed in aggressive human cancer cells and primary tumours [95], where it regulates a pro-oncogenic FA network. In PCa MGL expression is also increased and is part of a gene signature associated with epithelial-to-mesenchymal transition and stem-like properties. Most importantly, MGL inhibition impairs PCa aggressiveness [96]. Together, these enzymes break down TGAs, releasing FAs and glycerol that can be used for energy generation by FAO. However, the molecular mechanisms that control the balance between lipogenesis and lipolysis in PCa cells is largely unknown.

### 2.2. Alterations of Cholesterol Metabolism in PCa

Another important metabolic process in PCa is cholesterol metabolism. PCa tumours are dependent on steroid hormones such as testosterone, which is derived from cholesterol, through AR activation [97].

Cellular and systemic cholesterol concentrations are fine-tuned by the master transcriptional regulators governing cholesterol homeostasis, such as SREBP-2, liver X receptors (LXRs), and nuclear factor erythroid 2 related factor-1 (NRF1) [98]. Accumulation of cholesterol and cholesterol-derived oxysterols deactivates the SREBP-2 pathway, whereas hypoxia and nutrients depletion upregulate this pathway and its downstream targets.

Cellular cholesterol can either be imported from an extracellular source or endogenously synthesized de novo through the mevalonate pathway [98], which is upregulated in PCa [99]. The first step in cholesterol synthesis is catalysed by acyl-CoA: thiolase (AACT), which converts acetyl-CoA into acetoacetyl-CoA (Figure 2). This is followed by condensation to 3-hydroxy-3-methylglutaryl-coenzyme A (HMG-CoA) by the HMG-CoA synthase (HMGCS). Finally, HMG-CoA is reduced to mevalonate by HMG-CoA reductase (HMGCR), the main target of statins, and the rate limiting reaction of the cholesterol synthesis pathway. Several studies have suggested a potential benefit for cholesterol-lowering statin therapy in PCa. However, there are conflicting results as a recent preclinical study has suggested that statins may increase the aggressiveness of PCa [100].

Cells can also acquire cholesterol from low-density lipoprotein (LDL) taken up from the circulation via LDL receptor (LDLR)-mediated endocytosis [101]. However, in PCa tissue samples, low levels of LDLR expression have been identified in dedifferentiated tumours and in those with lethal outcomes, indicating a greater reliance on cholesterol synthesis than uptake [102]. As opposed to the LDLR, the high-density-lipoprotein receptor (SR-BI) is upregulated in PCa and correlates with a high Gleason grade and reduced disease-free survival in primary PCa [103]. In PCa, loss of PTEN through the activation of PI3K–Akt signalling leads to accumulation of cholesteryl esters (CE) by increasing cholesterol uptake and causing further esterification [104]. Interestingly, many epidemiological and preclinical studies have found that circulating cholesterol plays an important role in PCa progression, suggesting that hypercholesterolemia increases the risk of aggressive PCa and may be a risk factor for developing CRPC [99,105].

Cholesterol in excess is either exported from the cell by ATP-binding cassette (ABC) transporters [106], or converted to less toxic CEs by acyl-coenzyme A:cholesterol acyltransferases (ACATs) and then stored in LDs or secreted within lipoproteins. Tumour suppressor p53 upregulates the cholesterol-efflux transporter ABCA1 in many cancers, including PCa, thereby restricting SREBP2 maturation and subsequently repressing the mevalonate pathway [107].

### 2.3. Alterations of Phospholipid Metabolism in PCa

FAs are also used to synthesize phospholipids, which are essential components of cell membranes and key players in regulating signalling transduction and cell cycle [108]. In PCa, in particular, an important fraction of FAs are incorporated in phospholipids [109]. The most abundant phospholipids in membrane bilayers are phosphoglycerides, specifically phosphatidylcholine (PtdC) and phosphatidylethanolamine (PtdE), which are composed of two FAs esterified to a glycerol backbone and include choline and ethanolamine as head groups, respectively.

The metabolism of phospholipids is complex and highly altered in PCa, as reviewed in [36]; however, here we will focus on the metabolism of choline which is related with epigenetics. Activated choline metabolism is a hallmark of carcinogenesis and tumour progression, leading to elevated levels of PtdC in all types of cancer tested so far, including PCa [110], which is consistent with active membrane remodelling and cellular proliferation. Choline is an essential nutrient present in beef, eggs, wheat, and soybeans, whose uptake is increased in PCa by four types of choline transporters that mediate this process with different affinities for the substrate [111]. Diet is not the only source to obtain choline, but it can also be synthesized endogenously through the degradation of PtdC by phospholipases and this process seems to be upregulated in some cancer types such as PCa [110,111]. ADT may influence choline metabolism or choline uptake, but the evidence is not completely clear. Importantly, this flux of choline is used for diagnosis thanks to (11)C-choline and (18)F-choline PET/computed tomography (CT) in routine clinical practice, with impact on treatment decisions [112,113].

PtdC is produced from choline mainly through the Kennedy pathway, which uses acyl-CoA as building substrate [111,114]. This route has three main enzymes, choline kinase (CK), phosphocholine cytidylyltransferase (CCT), and cholinephosphotransferase 1 (CHPT1) (Figure 2). Increased expression and activity of CK has been reported in PCa, along with high levels of PtdC [115]. The expression of CCT is also of prognostic value in cancer [116]. CHPT1, which is responsible for the last step of de novo biosynthesis of PtdC, has been shown to be overexpressed in some tumours, and its overexpression positively correlates with tumour growth [117]. Focusing on PCa, data from public databases indicate a correlation between CHPT1 mRNA levels and PCa Gleason score and lymph node involvement [118]. Interestingly, androgen regulates CHPT1 gene transcription. In vitro and in vivo models of enzalutamide-resistant cells demonstrate that CHPT1 knockdown resensitizes enzalutamide-resistant cells to the drug. In addition, PtdC can be synthesized from PtdE by the phosphatidylethanolamine N-methyltransferase (PEMT) enzyme [111].

### 2.4. Targeting Metabolic Vulnerabilities in PCa

As previously described, lipid metabolism is a hallmark of PCa, implicated not only in its development but also in its pathological characteristics, aggressiveness, evolution and progression. Many of the metabolic pathways discussed are related to the AR pathway or to other genetic aberrations common in PCa. As we have seen, these alterations in lipid metabolism may be promising prognostic factors, and open a profoundly interesting field for the discovery of new therapeutic targets, with the potential of increasing the therapeutic armamentarium of PCa or synergizing with already approved therapies. However, it is important to note that, despite growing evidence of this close relationship, little progress has been made in translating all these breakthroughs to the clinic, and not many clinical trials in patients have been designed or are being developed (Table 1). Therefore, we must redouble our efforts to make the boom in metabolic therapies a reality in the coming years.

## 3. The Epigenome of PCa

Epigenetics refers to changes in gene function that are mitotically and/or meioticallyinheritable and do not entail a change in DNA sequence [123]. These changes occur in response not only to internal cellular factors but also to external factors; epigenetics can thus act as the link between the environment and the phenotype. Collectively, epigenetic mechanisms of gene regulation make up the epigenome, which consists of a complex network of interactions between DNA methylation, chemical histone modifications, histone variants, and non-coding RNAs, all of which are regulated by numerous enzymatic mechanisms [124,125,126,127,128]. These epigenetic marks do not change the DNA sequence, but instead modify chromatin structure and DNA accessibility, which influences numerous DNA-templated processes, such as gene expression and, ultimately, the phenotype. Accordingly, aberrant epigenetic marks translate into aberrant transcriptional programs that are characteristics of many diseases and are a nearly universal event in cancer [26,27].

PCa arises from both genetic and epigenetic alterations [129,130]. It seems that epigenetic changes occur earlier than genetic ones as DNA methylation alterations occur consistently in preneoplastic lesions [131]. However, epigenetic alterations are key not only to PCa initiation but also to PCa progression. Beltran and colleagues provided interesting insights into NEPC evolution as an adaptation from “classical” CRPC cells, rather than a linear or independent clonal evolution, and suggested that this lineage reprogramming is driven largely by the dysregulation of the epigenome and transcriptional networks [23]. In fact, aberrant DNA methylation patterns and altered expression of epigenetic modifiers such as enhancer of zeste 2 polycomb repressive Complex 2 Subunit (EZH2), transcription factors such as SRY-box transcription factor 2 (SOX2), brain-specific homeobox/POU domain protein 2 (BRN2) and N-Myc, and RNA-modifying factors are hallmarks of NEPC [23,132]. In particular, the epigenetic modulator EZH2, a master regulator of NEPC reprogramming [23,133], is frequently overexpressed in patients who have progressed to NEPC; consequently, its inhibition may prevent ARPI-mediated neuroendocrine differentiation [134,135].

### 3.1. Alterations of DNA Methylation in PCa

DNA methylation is the addition of a methyl group to cytosines within the dinucleotide CpG mediated by DNA methyltransferases (DNMTs), which use S-adenosylmethionine (SAM) as the methyl donor [136,137]. DNA methylation has traditionally been considered an inactive mark associated with closed chromatin and transcriptional silencing. However, this is not totally true; its function is highly genomic context-dependent, and DNA methylation in promoters and enhancers or in repeat sequences plays a repressive role, while DNA methylation affecting body genes often results in increased transcriptional activity [138].

Tumour cells are characterized by gains (hypermethylation) and losses (hypomethylation) of DNA methylation compared to normal tissue [26,27]. Hypermethylation is usually local and can occur in regulatory elements, where it causes the silencing of genes involved in multiple cellular processes, such as apoptosis, cell cycle, and DNA repair, many of which are tumour suppressor genes. In contrast, hypomethylation is often global and affects large genome domains; it is associated with chromosome instability, activation of transposable elements and pro-oncogenic genes, and loss of genomic imprinting.

DNA methylation has several characteristics which make it a promising source of biomarkers: it is stable, even in fixed samples, over time; it can be measured by simple and quantitative well-established methods; and it can be detected in different body fluids such as blood, urine, semen, saliva, etc. [139]. Several studies analysing the DNA methylome of PCa have found that DNA methylation patterns can distinguish benign prostate tissue, primary PCa, and mCRPC [140,141], and also differentiate between epigenetic subtypes of mCRPC [142]. Additionally, DNA methylation patterns have been associated with specific genetic alterations [143].

In early stages of PCa, DNA hypermethylation occurs in many genes, such as glutathione-S-transferase pi 1 (*GSTP1*, which belongs to the GST family of detoxification enzymes), *PTEN* (the most frequently lost tumour suppressor gene in Pca), and Ras association domain family protein 1, isoform A (*RASSF1A*), indicating that epigenetic alterations are early events in Pca [130,144,145,146,147]. Importantly, the *AR* promoter itself has also been found hypermethylated in a subset of CRPCs [148]. Some of these epigenetic alterations have been associated with different clinicopathological variables, suggesting that they may be potential biomarkers [145].

Though to a lesser extent than hypermethylation, gene promoter hypomethylation also occurs in PCa, for example in cancer/testis antigen gene (*CAGE*), cytochrome P450 family 1 subfamily B member 1 (*CYP1B1*), and heparinase (*HPSE*) [128,144,145,146,147]. As occurs with hypermethylation, the hypomethylation and consequent upregulation of some of these genes have been associated with clinical variables, suggesting their utility as biomarkers. Global hypomethylation, which affects repetitive sequences such as LINE-1 and Alu elements [149,150], has also been associated with PCa development and progression [151,152].

Not only is DNA methylation altered, but some of the enzymes involved in the DNA methylation/demethylation processes are also deregulated in PCa. DNMTs are overexpressed in PCa, whereas Ten-Elevated Translocation (TET) genes, involved in DNA demethylation, are underexpressed. Moreover, some of these genes are mutated in PCa with higher mutation rates in mCRPC [142]. Thus, the interest of DNA methylation modulators as therapeutic targets has grown in recent years, and different epigenetic compounds targeting these enzymes have been developed. However, demethylating agents, such as the DNMT inhibitor 5-azacytidine as well as 5-aza-2′-deoxycytidine, have been shown to be effective in the treatment of myelodysplastic syndromes [153] and have antitumour activities in in vitro experiments and animal models of several types of cancer, including PCa [154,155,156], but clinical trials have shown no significant effects [144]. Therefore, additional studies are required to assess the role of demethylating agents as therapeutic options for PCa, alone or in combination with other drugs.

Taken together, these data demonstrate the importance of DNA methylation in PCa both in tumour development and progression and for diagnosis and therapy.

### 3.2. Alterations of Histone Modifications in PCa

Chromatin is a DNA-protein complex whose basic repeating structural unit is the nucleosome, which consists of 147 bp of DNA wrapped around a histone octamer (two copies each of H2A, H2B, H3, and H4) [157]. Ultimately, chromatin can be further compacted to form higher-order structures restricting DNA access. Histones can be modified through the post-translational addition of different chemical groups (e.g., methyl, acetyl, phosphate, and ubiquitin) to specific amino acids by a broad range of enzymes, such as histone acetylases (HATs), histone deacetylates (HDACs), histone methyltransferases (HMTs) or histone demethyltransferases (HDMTs) [124]. Histone modifications are epigenetic marks that can regulate gene expression by altering chromatin and leading to a more open or a more compact structure, respectively associated with transcriptional activation or repression. These effects can be exerted either directly by upsetting the interaction between DNA and histones or indirectly through the recruitment of enzymatic chromatin modifiers. As a result, some histone marks are considered active, such as the acetylation of any lysine (H3K4Ac, H3K27Ac, and H4K20Ac) or the methylation of specific lysine residues (H3K4me, H3K36me, and H3K79me), while others are considered repressive, such as some lysine methylations (H3K9me and H3K27me).

Several studies have reported the alteration of global levels of histone modifications in PCa [158]. In localized PCa, H3K9me2/3, H3K4me1, AcH3, and AcH4 are reduced compared to benign prostate tissue, while H3K4me1/2/3 and H3K18Ac are increased in CRPC [159,160]. General hypomethylation of H4K20me1/2 in mCSPC and CRPC has also been reported [161]. Interestingly, some of these marks are associated with clinicopathological parameters: H3K4me1 correlates with PSA recurrence; H3K18Ac and H3K4me2 predict relapse-free survival; H4K20me1 correlates with lymph node metastases; and H4K20me2 correlates with Gleason score. Recent studies have also shown an increase in global H3K27me3 levels associated with PCa progression and metastasis and correlating with Gleason score [162,163].

Many histone-modifying enzymes have been shown to participate in prostate carcinogenesis [129,144,164]. For example, HDAC1/2/3 are highly expressed in CRPC [165], and the lysine specific demethylase 1 (LSD1) has been linked to risk of relapse [166]. The most studied epigenetic enzyme in PCa is EZH2, an HMT responsible for H3K27 trimethylation [167]. EZH2 is overexpressed in PCa, particularly in mCRPC, and has been strongly related to DNA hypermethylation [168]. It correlates with Gleason score and poor prognosis and is a master regulator of NEPC reprogramming.

Interestingly, androgen stimulation can trigger changes in histone patterns. For instance, AR recruits HATs and co-activators with HAT activity, such as p300 and CREB binding protein (CBP), to the promoter and enhancers of kallikrein 3 (KLK3) gene (encoding PSA) and other AR targets, promoting their transcriptional activation [169]. In contrast, HDMTs can also be recruited through androgen stimulation-dependent mechanisms, resulting in an inactive chromatin state [170], suggesting that histone acetylation levels of AR target genes have a key role in AR function.

In summary, like aberrant DNA methylation, alterations in histone modifications and histone-modifying enzymes are a hallmark of cancer and have a potential role both as biomarkers and as therapeutic targets [144,158]. The best studied histone modulators in PCa treatment are HDAC inhibitors, but outcomes of clinical trials are not as promising as those from preclinical studies [144], thus further research is warranted.

## 4. Influence of Diet on PCa, Lipid Metabolism and the Epigenome

Many epidemiological studies have observed that obesity is related to a higher incidence of high-risk or aggressive PCa and of PCa recurrence and, thus, an increased risk of PCa progression and death [171,172,173]. Increased visceral fat has been also associated with an increased risk of developing PCa [174]. Men with hypercholesterolemia are also at high risk of developing aggressive PCa [99]. Marin-Aguilera and colleagues recently demonstrated that hypercholesterolemia levels in mCRPC patients was associated with poorer clinical outcomes [105]. Dietary fat, as a fundamental contributor to obesity, may explain, at least partly, the complex link between obesity and PCa, as evidenced by the fact that countries with a higher dietary fat intake show higher PCa mortality rates [175]. However, although the association between fat intake and PCa risk has been extensively studied, it still remains controversial. Several studies point out to the importance of the composition of FAs rather than the total dietary fat intake. In line with this, some epidemiological studies have shown that high intake of animal and saturated fats is associated with increased risk of advanced or fatal PCa [176,177,178,179]. In contrast, vegetable fat intake has been associated with a lower risk of fatal PCa [180]. In addition to fats, other nutrients, which are not the focus of this review, are also related to PCa, as reviewed elsewhere [63,181]. Collectively, these findings raise the possibility that nutritional interventions may be useful for prevention of development of aggressive tumours in men at high risk of PCa and/or PCa patients. However, more studies are needed to better understand the complex relationship between diet and PCa.

One of the proposed mechanisms that could explain the association between HFD and PCa is through lipid metabolism [182]. Accordingly, the increased uptake of dietary lipids affects the formation of LD, as they are stored in form of LD in prostate cells, influencing lipid metabolism and contributing to the development of PCa [91]. Moreover, FASN expression in LNCaP (Lymph Node Carcinoma of the Prostate), a PCa cell line, xenograft mice fed with HFD was enhanced and correlated with PCa progression [183]. More importantly, a diet enriched in SFA promoted mCRPC in a PTEN knockout (KO) mouse model through an aberrant lipogenic program regulated by SREBP [55]. In the same line, a HFD enriched in SFA in another mouse PCa model enhanced a MYC proto-oncogene, bHLH transcription factor (MYC) transcriptional signature through metabolic alterations [184]. Switching from a HFD to a low-fat diet, the MYC transcriptional program was attenuated. Interestingly, SFA intake was also associated with a MYC transcriptional signature in PCa patients, which predicted PCa progression and death. Overall, these results point the influence of diet on lipid metabolism and open new avenues of treatment involving changes to the diet.

As a result of the influence of dietary components on metabolism, the availability of certain metabolites is affected, which in turn modifies epigenetic mechanisms because some metabolites are used by epigenetic enzymes as cofactors or act as direct epigenetic modulators, as summarized further below. These epigenetic alterations may regulate, at least partly, the transcriptional programs induced by diet. In line with this, the MYC transcriptional program enhanced by HFD in a murine PCa model was mediated by metabolic alterations that favoured the histone H4K20 hypomethylation at the promoter regions of MYC regulated genes, leading to increased cellular proliferation and tumour burden [184].

## 5. Interaction between Lipid Metabolism and Epigenetics in PCa

There is no doubt that metabolic and epigenetic alterations are hallmarks of cancer cells and are reciprocally linked [185,186,187]. The activity of some epigenetic enzymes depends on metabolic intermediates such as Acetyl-CoA, non-acetyl acyl-CoAs, NAD+, and S-adenosylmethionine (SAM), which act as substrates or cofactors, indicating that the epigenetic machinery is susceptible to metabolite fluctuations, which can occur in response to internal or external stimuli, including diet. At the same time, epigenetic alterations can drive metabolic reprogramming by influencing the expression of metabolic enzymes. As described above, many studies have reported an association between diet, obesity, and PCa development and progression, and epigenetics may be the bridge connecting these factors. Given that lipid metabolism is a key player in PCa, here we focus on the interaction between lipid metabolism and epigenetics, in particular DNA methylation and histone modifications.

### 5.1. Impact of Lipid Metabolism on Epigenetics

#### 5.1.1. FAs, Acetyl-CoA, and Histone Acetylation

The metabolite acetyl-CoA is a key factor in metabolism. It is derived from different sources, such as pyruvate, citrate, acetate, FAs, and amino acids, through different metabolic pathways. It is essential for a wide range of processes, such as ATP production, FA synthesis, steroid synthesis, and protein acetylation, including the epigenetic mark histone acetylation [188].

Histone lysine residues can undergo acetylation-deacetylation switches that are regulated by the action of HATs and HDACs, respectively. HATs catalyse the transfer of an acetyl group from acetyl-CoA to the lysine residues of histones, which neutralizes the lysine’s positive charge and decreases the ionic affinity between DNA and histones, promoting DNA accessibility. For this reason, histone acetylation is considered an active mark associated with open chromatin and transcriptional activation. In addition, histone acetylation acts to recruit numerous activating factors [189].

Histone acetylation levels have been associated with different clinical variables in PCa [159,160]. Therefore, a better understanding of how tumour histone acetylation levels are governed is clinically important. Although multiple factors are involved, several studies have found that histone acetylation is highly sensitive to the availability of acetyl-CoA, the universal substrate of HATs [190,191,192], and PCa is no exception, as demonstrated by the oncogenic activation of Akt that drives changes in acetyl-CoA production and histone acetylation in prostate tumours [193]. Interestingly, acetyl-CoA levels can not only alter the histone acetylation levels but also the pattern, as has been shown with p300, whose specificity can be perturbed by acetyl-CoA levels so that acetylation increases at some sites and decreases at others [194]. Although no specific studies have been performed in PCa in this regard, this seems to be a widespread mechanism. Thus, the different pathways regulating acetyl-CoA levels in PCa cells can influence histone acetylation levels and patterns and consequently modify the transcriptome and cellular phenotype.

The canonical pathway to generate acetyl-CoA occurs in the mitochondria, where the pyruvate generated by glycolysis is converted to acetyl-CoA by the pyruvate dehydrogenase complex (PDC) (Figure 2) [195]. PDC has also been found in the nucleus of PCa cells, where it controls the expression of SREBP target genes by mediating histone acetylation. Specifically, inactivation of PDC by inhibiting the subunit pyruvate dehydrogenase A1 (PDHA1) has been shown to inhibit PCa development in mouse and human xenograft tumour models, affecting H3K9 acetylation and the expression of lipogenic genes [196]. Interestingly, PDH1A and the PDC activator pyruvate dehydrogenase phosphatase 1 (PDP1) are amplified and overexpressed in PCa. These findings may pave the way for the targeting of the nuclear function of PDC in PCa patients, which could be achieved by using pyruvate analogs or alternatively by developing inhibitors of PDHA1.

Under starved conditions (low glucose levels), the mitochondrial acetyl-CoA can be generated by FAO from free FAs (Figure 2). This is especially important in PCa, which, unlike other cancer types, relies on FAO as the dominant energetic pathway, mostly in early disease stages [80]. Alternatively, acetyl-CoA can also be obtained from the short chain FA (SCFA) acetate in the cytosol by the acyl-CoA synthetase short chain family member 2 (ACSS2). ACSS2 is also located in the nucleus of some cancer cells, including PCa [197], where its main function is to maintain histone acetylation under conditions of limited oxygen and nutrient availability by recapturing the acetate released as a consequence of histone deacetylation reactions.

Under fed conditions (high glucose levels), the mitochondrial citrate derived from acetyl-CoA can be transported to the cytosol or to the nucleus and converted to cytosolic or nuclear acetyl-CoA by ACLY. Cytosolic and nuclear acetyl-CoA are then used for FA synthesis or histone acetylation, respectively [198]. Interestingly, Lee et al. reported that the oncogene Akt can regulate the levels of histone acetylation by enhancing acetyl-CoA synthesis through the phosphorylation of ACLY, even during nutrient limitation, in PCa [193].

Although cytosolic acetyl-CoA can enter the nucleus and influence the nuclear acetyl-CoA pool—and vice versa—it seems that nuclear production is mainly regulated by nuclear ACLY, ACSS2, or PDH. Therefore, glucose and FA availability, as well as subcellular compartmentalisation of some metabolites, can affect acetyl-CoA levels and, consequently, histone acetylation.

Intriguingly, the chemical properties of acetyl-CoA allow the spontaneous acetylation of lysines and, as a result, the local generation of acetyl-CoA can regulate protein acetylation, including that of histones [199]. Therefore, the nuclear metabolic enzymes ACLY, ACSS2, and PDH that regulate nuclear acetyl-CoA levels can act as direct epigenetic regulators [192].

The levels of acetyl-CoA can be also indirectly influenced by arginine in PCa. Argininosuccinatesynthetase (ASS), the enzyme responsible for intracellular arginine synthesis, is repressed in some tumour types, including PCa [200]. ASS-deficient tumours require external arginine for growth and survival. In these tumours, arginine acts as an epigenetic regulator that induces global histone acetylation through the activation of the mTOR pathway, which upregulates ACLY and ACSS2 and consequently increases acetyl-CoA levels [197].

Another promising finding related to Acetyl-CoA levels from a therapeutic point of view derives from the study by Xu Q et al. [201]. They showed that chemoresistant PCa cells were sensitive to HDAC inhibitors, specifically Trichostatin A (TSA) and suberoylanilide hydroxamic acid (SAHA), and those cells with higher levels of acetyl-CoA and hyperacetylated proteins, including histones, were more sensitive. Thus, the levels of acetylation proteins and/or nuclear-cytosolic acetyl-CoA may be potential biomarkers to predict HDAC inhibitors response in chemoresistant PCa cells.

#### 5.1.2. SCFAs, Non-Acetyl Acyl-CoAs, and Histone Acylation

Acetylation was the first identified histone acyl modification. However, as a result of advances in high-sensitivity mass spectrometry techniques, the catalogue of known histone modifications has greatly expanded to include new types of histone short-chain acylations (e.g., butyrylation, crotonylation, and malonylation) [202]. Histone acylation differs from histone acetylation in the length of the carbon chain and in the charge. A wealth of evidence indicates that histone acylation can affect gene regulation and is functionally distinguishable from histone acetylation [203]. No specific enzymes for these histone marks have been identified; rather, the different acyl groups can be added by previously identified HATs. It is assumed that the levels of a specific histone acylation would depend on the relative levels of the corresponding acyl-CoA and the rest of the acyl-CoAs, which would compete for specific HATs [204]. In turn, the levels of non-acetyl acyl-CoAs depend on the levels of SCFAs, most of which are generated by intestinal microbial anaerobic fermentation of dietary fibre, indicating that histone acylation levels are largely dependent on diet [205]. Adding SCFAs, such as crotonate or β-hydroxybutyrate, to the cell culture medium increases cellular concentrations of their respective acyl-CoAs and histone acylation, suggesting that the SCFA is converted into its corresponding acyl-CoA, which is then used directly as a co-factor in a histone acylation reaction [202,206]. Therefore, SCFAs can act as epigenetic regulators by directly influencing histone acylation. Interestingly, in addition to competing for HATs, the different histone acylations may compete for other metabolic enzymes. For example, ACSS2 has been shown to be involved in the synthesis of crotonyl-CoA from the SFCA crotonate, although it is not known if the relationship is direct or indirect [207].

Little is known about the role of these histone marks and their relationship with lipid metabolism and diet in PCa. A recent study showed that the levels of histone crotonylation in PCa tissue were higher than in adjacent normal tissue and, moreover, increased with disease progression. In addition, in both androgen-dependent and castration-resistant PCa cell lines, histone crotonylation was able to specifically activate the AR pathway and promote cell proliferation, migration, and invasion [208]. This epigenetic mark may thus be a potential prognostic biomarker and therapeutic target.

#### 5.1.3. SCFAs and Histone Deacetylases

SCFAs can also influence the chromatin structure by inhibiting HDACs [205]. Studies have revealed that among the SCFAs, butyrate is the most effective HDAC inhibitor, followed by propionate. Butyrate has multiple effects in PCa, including induction of differentiation, growth arrest, and induction of apoptosis, which are partly due to its capacity to inhibit HDAC activity [209,210,211]. Accordingly, treating cells with butyrate results in histone hyperacetylation and thus remodelling of chromatin towards an open and transcriptionally competent state. Importantly, Paskova et al. showed in in vitro models that butyrate had an effect on PCa cells, but not on normal cells; specifically, it decreased cell viability, induced AR coregulators expression, and activated transcription activity, at least partly through the increase in H4K18Ac and H4K12Ac [212].

These findings may be important for potential incorporation of butyrate, specifically sodium butyrate, into PCa therapy. Moreover, the manipulation of butyrate by alterations in the microbiota through diets that favour the gut generation of butyrate can be potentially taken into consideration for PCa treatment/prevention.

#### 5.1.4. FAs and Sirtuins

Sirtuins (SIRT1 to SIRT7), initially classified as class III nicotinamide adenine dinucleotide (NAD+)-dependent HDACs [213,214,215], are now known to be localized in different subcellular compartments (nucleus, cytosol, and mitochondrion), where they have specific substrates and functions, including response to stress, genome stability, aging, and metabolism homeostasis. Sirtuins deacetylate not only histones but also non-histone proteins. Additionally, they have deacylase activity and can remove other single acyl groups, such as succinyl, malonyl, and even long-chain fatty acyl groups [216,217]. Sirtuins are connected to lipid metabolism at different levels and bidirectionally. The most direct link is by mithocondrial sirtuins, which can deacetylate metabolic enzymes; for example, SIRT3 deacetylates ACO2 in PCa [218]. However, here we focus on the sirtuins that act as epigenetic regulators, specifically SIRT1, SIRT2, SIRT6, and SIRT7, which are found in the nucleus and can deacetylate specific lysine residues of histones (thus causing compaction of chromatin and gene inactivation), histone-modifying enzymes, and structural and transcription factors [219].

Sirtuins are NAD+-dependent, and a widely accepted hypothesis is that fluctuations of NAD+ levels can regulate sirtuin activity [214,220]. For instance, reduced intracellular NAD+ limits the deacetylase activity of SIRT1, resulting in elevated H4K16Ac levels [221]. The NAD+ homeostasis is maintained by biosynthesis, consumption and recycling in the different subcellular compartments—mitochondria and cytosol/nucleus. The principle source of NAD+ is from salvage pathways that recycle other adenine nucleotide metabolites, such as nicotinamide (NAM) [222]. NAD+ levels are also regulated through conversion to its reduced form NADH and vice versa. A mechanism of recycling NADH to NAD+ that is related to lipid metabolism is the reaction of desaturation of PUFAs catalysed by FADS1 and FADS1 [223], whose expression is high in PCa cells [224].

This ratio can be also influenced by diet. During calorie restriction, NAD+ levels increase and lead to the activation of some sirtuins, while with a HFD, the ratio NAD+/NADH decreases and leads to decreased activity of some sirtuins [225]. However, these dietary interventions seem to affect sirtuins differently in different tissues. No specific studies have explored this issue in prostate tissue or in PCa, but the fact that a HFD is related to PCa progression suggests a possible link between diet, sirtuins, and PCa that merits investigation.

SIRT6 also mediates a link between sirtuins and lipid metabolism. SIRT6 is almost exclusively localized in the chromatin, where it deacetylates H3K9Ac, H3K918Ac, and H3K56Ac, promoting transcriptional silencing [226]. Biologically relevant LCFAs, such as linoleic, oleic, and myristic, have recently been reported to enhance SIRT6 activity by inducing conformational changes that increase the binding affinity with acetylated H3 [227]. Additionally, nitrated FA can activate SIRT6 and promote H3K9Ac deacetylation [228]. Given these findings, we can speculate that SIRT6 may be activated in nutrient conditions where free FAs are increased, for instance, due to diet or fasting. Although regulation at the molecular level of other sirtuins by FAs has not been investigated, several clinical trials have evaluated the effect of dietary FAs on SIRT1 expression and activity [229]. The evidence from these studies is insufficient to understand how lipid consumption modulates sirtuins in humans; however, an appealing hypothesis cites oleic acid as a natural activator of SIRT1.

All together, these data suggest that sirtuins are potential predictive biomarkers and therapeutic targets in PCa. However, how sirtuins can be regulated through manipulating NAD+ levels in PCa requires further investigation in this direction.

#### 5.1.5. Phospholipids, SAM, and Histone Methylation

SAM is a universal methyl donor for all cellular reactions involving a methylation step catalysed by a methyltransferase that transfers a methyl group to a variety of substrates—including proteins, nucleic acids, lipids, and secondary metabolites—and releases the by-product S-adenosylhomocysteine (SAH) [230]. SAM is thus a common cofactor between lipid metabolism, specifically phospholipids, and histone methylation. The methylation of one of the major phospholipids, PtdE, to generate another phospholipid, PtdC, is the major SAM-consuming reaction and compete with DNMTs and HMTs for the use of SAM [231]. Cells deficient in PtdE methylation show increased SAM levels, and as a result, there is an aberrant hypermethylation of histones, indicating the dependence of histone methylation on SAM concentration. Another link between SAM, DNA/histone methylation and phospholipids is that choline, which can be synthesized from PtdC, is oxidized to betaine, which is a key donor of methyl groups to the homocysteine–methionine (HCys–Met) cycle. Perturbations in choline metabolism, which are common in PCa [111] can affect the HCys–Met cycle and thus the levels of SAM and the methylation of DNA and histones.

#### 5.1.6. Nuclear Lipogenic Enzymes

In addition to ACLY, ACSS2 and PDH, FASN can also be found in the nucleus of PCa cells [232]. Although nuclear FASN correlates with Gleason grade, the function of nuclear FASN is unknown. The protein contains an acetyltransferase domain, which is used in the synthesis of FA, and crystal structure has shown a conserved methyltransferase domain, whose function has not yet been demonstrated. Nevertheless, these findings suggest the potential of FASN as an epigenetic regulator through the acetylation and/or methylation of chromatin.

### 5.2. Impact of Epigenetics on Lipid Metabolism

Epigenetics can remodel cellular metabolism so that the aberrant expression of some lipid metabolism genes in PCa is partly due to epigenetic alterations (Figure 2). This is the case for FASN and ACC, key enzymes in de novo FA synthesis, which are hypomethylated in PCa compared to normal tissue, leading to high gene expression [233]. A potential lipogenic gene affected by DNA methylation is the desaturase SCD1, whose expression is frequently lost in PCa. As the low expression of SCD1 in glioblastoma has been related to DNA hypermethylation [234], it would be interesting to study DNA methylation levels of SCD1 in PCa. Many reports have also shown that the elongase ELOVL2 suffers age-related DNA methylation changes, specifically hypermethylation, which, unlike most age-related DNA methylation changes, are not tissue-specific and thus can occur in prostate [235]. The hypermethylation of ELOVL2 is associated with decreased gene expression and impaired lipid metabolism [236].

Another gene regulated by DNA methylation is the FA transporter CD36, which presents different promoters and distal enhancers that can be hypermethylated in different contexts. Most CD36 DNA methylation data derive from genome-wide studies focused on metabolic diseases such as obesity [237], and only one study focused on cancer reported the correlation of CD36 DNA hypermethylation with progression in lung tumours [238]. Given the key role of CD36 in PCa, further investigation is needed.

Cholesterol metabolism is also influenced by DNA methylation. The loss of cytochrome P450 family 27 subfamily A member 1 (CYP27A1), which catalyses the rate-limiting hydroxylation of cholesterol to bile acid, has been reported to contribute to the dysregulation of cholesterol homeostasis in PCa [239] and to be associated with shorter disease-free survival and higher tumour grade. In the same study, the authors found a negative correlation between CYP27A1 expression and promoter DNA methylation, suggesting that this epigenetic mechanism may be involved in the silencing of CYP27A1 in PCa. ABCA1, the major cellular cholesterol efflux transporter, has also been found hypermethylated in PCa, specifically in intermediate- to high-grade tumours. Remarkably, ABCA1 expression levels are inversely correlated with Gleason grade [240].

Although the expression of many of these genes is controlled by the master transcription factors SREBPs, which are overexpressed in PCa [241], these transcription factors recruit epigenetic factors to enhance gene activation. In hepatocytes, SREBP1 and SREBP2 recruit Brahma-related gene 1 (Brg1), a chromatin remodelling protein, to target gene promoters, such as FASN and ACC, trans-activating their gene expression, and this recruitment parallels concomitant H3 acetylation [242,243]. Consistent with these findings, in PCa cells, Brg1 is recruited to the ELOVL3 promoter by the retinoic acid receptor-related orphan receptor (RORγ) and interacts with the acetyltransferase p300, promoting H3K27Ac to activate ELOVL3 expression [244]. As ELOVL3 can also be activated by SREBP1, it is possible that the same mechanism occurs in prostate cells. It is also possible that lipogenic genes other than ELOVL3 could be targets of Brg1 and p300 in PCa. In fact, FASN is activated by p300, which enhances H3K27Ac [244]. Interestingly, evidence indicates that Brg1 can act as promoter of PCa oncogenesis [245,246], and further research is warranted to better understand the link between Brg1, histone acetylation, and lipid metabolism in PCa, as well as their potential as biomarkers and therapeutic targets.

Another epigenetic mechanism observed in different cancer types, including PCa, that activates genes involved in lipid metabolism occurs specifically under hypoxia, with the acetate-mediated increase in histone acetylation levels. Acetate can function as an epigenetic metabolite that induces H3 acetylation of H3K9, H3K27, and H3K56 at FASN and ACC promoters, which upregulates FASN and ACC expression and increases de novo lipid synthesis to promote tumour cell survival [247].

The repression of some lipid metabolism genes is mediated by HDACs, including sirtuins. Bile acids stimulate the sequential recruitment of HDAC7, 3, and 1, as well as the corepressors silencing mediator of retinoid and thyroid receptors-α (SMRTα) and nuclear receptor corepressor 1 (NCoR1) into the nucleus, forming a repressive complex on the CYP7A1 promoter that leads to its repression [248]. SIRT6 also regulates cholesterol metabolism. Specifically, SIRT6 can be recruited to the SREPB2 promoter, where it deacetylates H3, inhibiting its expression [249]. It is not known if this mechanism is specific to hepatocytes or occurs in other cell types; however, it is indirectly related to PCa as it influences the homeostasis of circulating cholesterol.

## 6. Concluding Remarks and Future Directions

Numerous studies on lipid metabolism and epigenetics in PCa have improved our understanding of PCa progression and therapy resistance. Importantly, different findings have revealed a bidirectional crosstalk between aberrant lipid metabolism and epigenetics as well as the influence of diet. However, we still do not have a complete picture of these complex connections requiring to be interpreted through a systems biology perspective in order to gain full insight into this matter. For example, what are the specific action and mechanism of the multiple acylation modifications on histones in PCa? Additionally, how can these histone acylation modifications be modulated by diet? There is still a great deal of research to be investigated. With better knowledge there is the possibility of creating better diagnostic testing tools and treatment options. The studies reviewed here pose that alterations of lipid metabolism and epigenetic marks may be established as prognostic and predictive factors; however, well-designed case–control studies and prospective trials are required to evaluate them. There is also a need to clearly define the mechanisms underlying the crosstalk between lipid metabolism and epigenetics in PCa that can be exploited therapeutically. It is time for the exciting opportunity to evaluate the modulation of cancer metabolism, through metabolic, epigenetic, and/or dietary intervention, as a strategy to improve the management of PCa.

## Figures and Tables

**Figure 1 nutrients-14-00851-f001:**
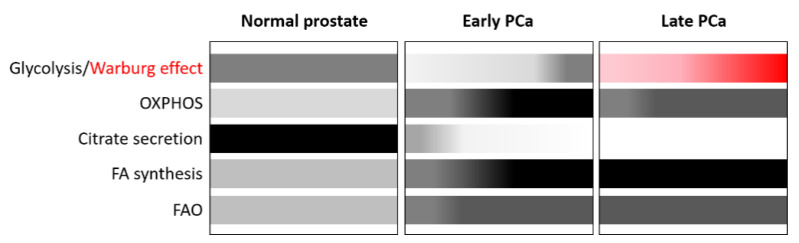
Metabolism in prostate cancer (PCa). Schematic representation of metabolic reprogramming in PCa. Normal prostate has a unique metabolic profile and is characterized by high rates of aerobic glycolysis and low rates of oxidative phosphorylation (OXPHOS). The tricarboxylic acid (TCA) cycle is truncated by zinc, whose uptake is favoured by androgens, and the enzyme aconitase 2 required for the conversion of citrate to isocitrate is inhibited allowing the secretion of high levels of citrate in the seminal liquid. Citrate can also be used for lipid biosynthesis. During malignant transformation, zinc is no longer accumulated and PCa cells reactivate the TCA cycle to oxidize citrate to produce energy and anabolic substrates such as acetyl-CoA. Instead of glucose, fatty acids (FAs) are the major bioenergetic source that feed TCA. Thus, there is an increase in the rates of OXPHOS while the rates of glycolysis decrease. De novo synthesis of FA and FA oxidation (FAO) are also increased in early PCa through the upregulation of some of the involved enzymes regulated by AR. In late stages of the disease, which are usually metastatic, there is an increased aerobic glycolysis, and the Warburg effect is observed, without decrease in OXPHOS. Both de novo lipogenesis and FAO also show high rates to fulfil the energetic and anabolic needs of cancer cells. The intensity of the colours (grey or red scale) indicates the levels of activity of each pathway.

**Figure 2 nutrients-14-00851-f002:**
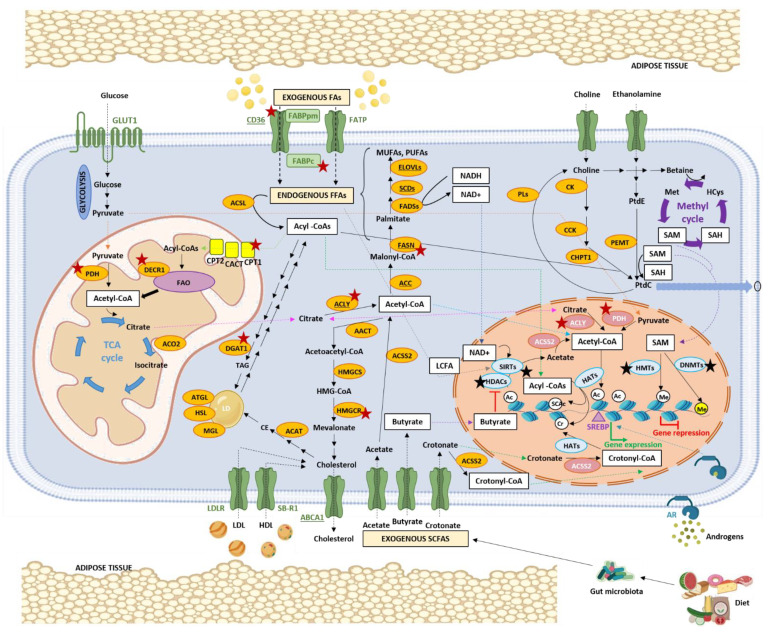
Schematic representation of lipid metabolism and the link with epigenetic marks in PCa. PCa cells show an increased uptake of exogenous FAs, which are transported by membrane-bound transporters. PCa cells also have increased de novo FA synthesis. Additionally, intracellular FAs can mobilize via lipid droplet lipolysis. In the cytosol, FAs can either be stored as TAG in lipid droplets or undergo enzymatic conversion to acyl-CoAs, which can enter the mitochondria via the carnitine shuttle system. Inside the mitochondria, acyl-CoAs are broken down through a series of enzymatic reactions known as FAO. The resulting acetyl-CoA enters the TCA cycle, where it is oxidized for citrate production. Citrate can be transported to the cytosol, where it is converted to acetyl-CoA, which can then be used to synthesize cholesterol by the mevalonate pathway or phospholipids. Exogenous cholesterol can also be obtained as lipoproteins through the receptors LDLR and SB-R1. All these processes generate metabolic intermediates, some of which (white boxes) act as substrates or cofactors of epigenetic enzymes. Specifically, acetyl-CoA is used by HATs to acetylate histones; acyl-CoAs are used by HATs to acylate histones; SIRTs are dependent on NAD+; SAM is used by HMTs and DNMTs to methylate histones and DNA, respectively. Additionally, some SCFAs, such as butyrate, can inhibit HDACs, while some LCFAs can enhance sirtuin activity. Dotted lines correspond to metabolites that are transported or diffused through membranes (a different coloured line for the each metabolite). Underlined enzymes correspond to enzymes whose expression can be epigenetically regulated. Enzymes marked with a star (red for metabolic enzymes and black for epigenetic enzymes) are enzymes whose inhibition has an anti-oncogenic effect in PCa. Abbreviations: FA, fatty acid; FFA, free FAGLUT1, glucose transporter 1; PDH, pyruvate dehydrogenase; TCA, tricarboxylic acid cycle; ACO2, aconitase 2; DECR1, 2,4-dienoyl-CoA Reductase 1; FAO, FA b-oxidation; CPT/2, carnitine palmitoyltransferase 1/2; CACT, carnitine/acylcarnitine translocase; FATPs, FA transport proteins; FABPpm, plasma membrane FA binding protein; FABPc, cytosolic FABP; ACSL, acyl-CoA synthetase long chain family member; DGAT1, diacylglycerol O-acyltransferase 1; TAG, triglyceride; ATGL, adipose triglyceride lipase; HSL, hormone-sensitive lipase; MGL, monoglyceride lipase; LD, lipid droplet; LDL, low density lipoprotein; HDL, high density lipoprotein; LDLR, LDL receptor; SB-R1, scavenger receptor class B member 1; ABCA1, ATP binding cassette subfamily A member 1; HMGCR, 3-hydroxy-3-methylglutaryl-CoA reductase; HMGCS, 3-hydroxy-3-methylglutaryl-CoA synthase 1; AACT, acyl-CoA: thiolase; ACAT, acyl-coenzyme A:cholesterol acyltransferase 1; CE, cholesteryl ester; ACLY, ATP–citrate lyase; ACSS2, acyl-CoA synthetase short-chain family member 2; ACC, acetyl-CoA carboxylase; FASN, FA synthase; MUFAs, monounsaturated FAs; PUFAs, polyunsaturated FAs; SCDs, stearoyl-CoA desaturases; FADSs, FA desaturases; ELOVLs, elongation of very long-chain fatty acid protein; LCFA, long chain FA; SCFA, short chain FA; PtdE, phosphatidylethanolamine; PtdC, phosphatidylcholine; CK, choline kinase; CCK, cholecystokinin; CHPT1, choline phosphotransferase 1; PEMT, phosphatidylethanolamine N-methyltransferase; PLs, phospholipases; Met, methionine; HCys, homocysteine; SAM, S-Adenosylmethionine; SAH, S-adenosylhomocysteine; DNMTs, DNA methyltransferases; HATs, histone acetylases; HDAC, histone deacetylases; HMT, histone methylases; SIRTs, sirtuins; Ac, acetyl group; Me; methyl group; SCAc, short chain acyl group; Cro, crotonyl group; SERBFs, sterol regulatory element-binding proteins; and AR, androgen receptor. Created with Biorender (biorender.com, accessed on 10 December 2022).

**Table 1 nutrients-14-00851-t001:** Overview of clinical trials with drugs targeting lipid metabolism.

Pathway	Focus	Drug	Disease	Phase	Patients	Objective	Status	Results	Idenitifier
Cholesterol	HMGCR	Rosuvastatin	Metastatic PCa	Phase 4	70	Agressive parameters	Completed	Not published	NCT04776889
		Atorvastatin	Localized PCa	Phase 2	160	Agressive parameters	Completed	[119]	NCT01821404
			Localized PCa	Phase 2	354	Recurrence rate	Completed	[120]	NCT01759836
			Metastatic PCa	Phase 3	400	Recurrence rate	Recruiting	Not published	NCT04026230
		Atorvastatin + celecoxib	Localized PCa	Phase 2	27	PSA response	Completed	Not published	NCT01220973
		Atorvastatin + AAS Acetylsalicylic Acid	Castration Resistant	Phase 3	1210	Overall Survival	Recruiting	Not published	NCT03819101
		Simvastatin	Localizaed PCa	WOP	42	Changes in Mevalonate Pathway	Completed	Not published	NCT00572468
		Simvastatin + Ezetimibe	Localized PCa	WOP	63	Agressive parameters	Completed	Not published	NCT02534376
		Fluvastatin + Pimonidazole	Localized PCa	WOP	33	Agressive parameters	Completed	[121]	NCT01992042
Fatty acid	FASN	Omeprazole	Metastatic PCa	Phase 2	20	Response rate	Recruiting	Not published	NCT04337580
		TVB-2640	Metastatic Solid tumour *	Phase 1	180	MTD	Completed	[122]	NCT02223247
	ACSS2	MTB-9655	Metastatic Solid tumour *	Phase 1	30	MTD	Recruiting	Not published	NCT04990739
	LXR	RGX-104	Metastatic Solid tumour *	Phase 1	135	MTD	Recruiting	Not published	NCT02922764
	OXPHOS	IACS-010759	Metastatic Solid tumour *	Phase 1	29	MTD	Completed	JCO2019_37:15_sup	NCT03291938
	CD36	VT1021	Metastatic Solid tumour *	Phase 1	116	MTD	Active	Not published	NCT03364400
		CVX-045	Metastatic Solid tumour *	Phase 1	40	MTD	Completed	Not published	NCT00879554
		ABT-510	Metastatic Solid tumour *	Phase 1	45	MTD	Completed	Not published	NCT00586092
	LDLR	ANG1005	Metastatic Solid tumour *	Phase 1	56	MTD	Completed	JCO2014 32:15_sup	NCT00539383

Abbreviations: HMGCR, 3-hidroxi-3-metil-glutaril-CoA reductase; PCa, prostate cancer; PSA, prostate-specific antigen; WOP, window of opportunity trial; FASN, fatty acid synthase; MTD, maximum tolerated dose; ACSS2, acyl-CoA synthetase short chain family member 2; LXR, liver X receptor; OXPHOS, mitochondrial oxidative phosphorylation system; and LDLR, low density lipoprotein receptor; AAS, Acetylsalicylic Acid. * Patients with PCa are included.

## Data Availability

No new data were created and analysed in this manuscript. Data sharing is not applicable.

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
