# Peer review of "Lipid Metabolism and Epigenetics Crosstalk in Prostate Cancer"

_nutrients, 2022, doi:10.3390/nu14040851_

Round 1

Reviewer 1 Report

I appreciated this interesting paper about the relation between epigenetics and lipid metabolism in prostate cancer. The originality of this review paper is the investigation of the interections between lipid meabolism and epigenetics , in particular  DNA methilation and histone modifications.

Reviewer 2 Report

Brief Summary: The aim of the study by Pardo et al. was to provide a comprehensive analysis of the complex link between lipid metabolism and epigenetic regulation in prostate cancer progression. More specifically, the authors discuss the importance of both de novo fatty acid (FA) metabolism as well as exogenous FA uptake in aggressive or neuroendocrine (NEPC) prostate cancer, and its potential implications on regulating distinct epigenetic processes during disease progression or therapy resistance.  Of particular interest is that the authors also briefly discuss the impact of prominent prostate cancer epigenetic alterations on lipid metabolism as well, providing speculations of how these may be functionally relevant for therapeutic targeting. This manuscript is timely, well written and could be of interest in a broad reader audience, suited for the special issue of “Current Understandings on the Role of Diet and Epigenetics in Cancer”. However, it would benefit from some minor additional clarifications or additions.

Comments:

  • The role of diet is not very clear throughout the manuscript. The authors may want to clarify how specific dietary aspects influence prostate cancer metabolism and in turn the epigenome or vice versa. They briefly touch upon this when the mention the exogenous FA synthesis and cholesterol metabolism.
  • The authors claim (based on published data) that early stages or pre-neoplastic lesions of prostate cancer are solely depended on aerobic glycolysis and not mitochondrial oxidative phosphorylation (OXPHOS). This is not entirely true as recent studies (i.e. PMID:33893149) contradict this paradigm and suggest that under certain genetic (NKX3.1 gene) and environmental conditions (oxidative stress) can induce mitochondrial OXPHOS dysfunction, promote aerobic glycolysis and drive a more aggressive early prostate cancer phenotype. The authors should discuss this aspect in the introduction since it may open new avenues to target subsets of prostate cancer patients with specific molecular and metabolic phenotypes.
  • A summary of past and current clinical trial dealing with aspects of lipid metabolic targeting in prostate cancer could be beneficial (but not necessary) to improve the clinical aspect of the review.

Reviewer 3 Report

This is a very wide-ranging and unfocused review of the role of epigenetics and lipid metabolism in prostate cancer.   Much of the review does not deal with prostate cancer directly and is more like a general review of epigenetics and lipid metabolism and of their possible role in cancer in general.   Therefore, it does not help the reader to sort out what is related to prostate cancer and what is not.   Furthermore, it frequently references other reviews rather than primary publications which is also not helpful to the reader. And not all references are correct. For example reference 158 does not deal with GSTP1 silencing as the authors claim on page 11 of the paper. 

Importantly, the authors do not really provide any detailed information about the role of diet and dietary fat in the risk and development of prostate cancer. And the paper does not critically evaluate many of the studies covered. For example, there is not as much certainty about the role of GSTP1 in prostate cancer as the authors and others claim. This lack of critical evaluation perpetuates the hypotheses proposed by others as facts without dealing with weaknesses versus strengths of the studies described in the paper. 

The sheer magnitude of information included in this paper without an attempt to reduce it to one or more manageable and understandable hypotheses is at the core of the lack focus and results in conclusions that are vague and not really informative.   The authors may want to consider separating the information in this review into a few more focused and less dense mini-reviews. 

Round 2

Reviewer 3 Report

The authors have responded to the previous criticisms with adding more information which makes this paper even more unwieldy.  They provide a reasonable rationale for how the paper was constructed, but they do not address the issue of lack of focus.

The addition of a section dealing with human dietary studies is in principle a good idea, but the table that is included does not provide much information.  It is rather useless to list studies that are still ongoing or have been terminated or completed without yielding any results.  The studies that have been completed are mostly not dealing with dietary fat specifically.  In about 20 minutes this reviewer was able to review those studies on clnicaltrials.gov and found the following information:

NCT00003367 dealt with an intervention with vitamin E, soy, and green tea, but not fat at all, for 4 years but no effect was found on the primary endpoint, PSA.  This study actually includes an agent (vitamin E) that has been shown to be harmful by increasing risk of prostate cancer and of diabetes. (Shike M et al. Lack of effect of a low-fat, high-fruit, -vegetable, and -fiber diet on serum prostate-specific antigen of men without prostate cancer: results from a randomized trial. J Clin Oncol. 2002;20(17):3592-8)

NCT00082732 studied men with hormone refractory prostate cancer counseled to consume a low-fat, high-fiber diet that includes soy protein for 52 weeks. The study was completed in 2006, but no results have been reported suggesting that the outcome was negative.

NCT01238172 was a study of telephone-based dietary counseling and structured dietary education to increase vegetable intake but not focused on fat in men with prostate cancer. The primary endpoint was time to progression which was not modified by the intervention (reference 182).

NCT00458549 was a study with 28 days of omega-3 fatty acids consumption prior to surgery which was terminated with no publications and no listed results

NCT02454517 studied the effects of Diabetes Prevention Program (DPP) lifestyle intervention, but was terminated because of lack of funding without any results.

NCT00475982 studied effects of an 8 week diet and exercise weight loss program followed by radical prostatectomy. No effects on prostate cancer parameters was found; the only effect was on weight loss (ref 183)

The bottom line is that there is not much evidence of a beneficial effect of dietary fat reduction on prostate cancer.  And the results of animal studies have been mixed and some studies used incorrect diets.  The epidemiology of fat intake and prostate cancer risk is not described in any critical detail in this review, but it is fair to state that the results of most recent studies have not been very supportive of an association between dietary fat as such and prostate cancer risk. This undermines the premise of this review. Perhaps this paper could be more focused and hypothetical rather than providing findings that are not very strong.
